# The structures of secretory and dimeric immunoglobulin A

Sonya Kumar Bharathkar[1†], Benjamin W Parker[1†], Andrey G Malyutin[2,3],
Nandan Haloi[4,5], Kathryn E Huey-Tubman[2], Emad Tajkhorshid[1,4,5],
Beth M Stadtmueller[1]*

[1]Department of Biochemistry, University of Illinois Urbana-Champaign, Urbana,
United States; [2]Division of Biology and Biological Engineering, California Institute of
Technology, Pasadena, United States; [3]Beckman Institute, California Institute of
Technology, Pasadena, United States; [4]Center for Biophysics and Quantitative
Biology, University of Illinois at Urbana–Champaign, Urbana, United States; [5]NIH
Center for Macromolecular Modeling and Bioinformatics, Beckman Institute for
Advanced Science and Technology, Urbana, United States

**Abstract** Secretory (S) Immunoglobulin (Ig) A is the predominant mucosal antibody, which binds
pathogens and commensal microbes. SIgA is a polymeric antibody, typically containing two copies
of IgA that assemble with one joining-chain (JC) to form dimeric (d) IgA that is bound by the
polymeric Ig-receptor ectodomain, called secretory component (SC). Here, we report the cryo-
electron microscopy structures of murine SIgA and dIgA. Structures reveal two IgAs conjoined
through four heavy-chain tailpieces and the JC that together form a β-sandwich-like fold. The two
IgAs are bent and tilted with respect to each other, forming distinct concave and convex surfaces.
In SIgA, SC is bound to one face, asymmetrically contacting both IgAs and JC. The bent and tilted
arrangement of complex components limits the possible positions of both sets of antigen-binding
fragments (Fabs) and preserves steric accessibility to receptor-binding sites, likely influencing
antigen binding and effector functions.

**\*For correspondence:**
bethms@illinois.edu

[†]These authors contributed
equally to this work

**Competing interests:** The
authors declare that no
competing interests exist.

**Reviewing editor:** Sjors HW
Scheres, MRC Laboratory of
Molecular Biology, United
Kingdom

## Introduction

The vertebrate mucosa is a vast extracellular environment that mediates host interactions with a
broad range of antigens including toxins, pathogens, and commensal organisms. The diversity of
these antigens, some of which are beneficial to the host and some of which are harmful, has driven
complex evolutionary interplay between mucosal immune system molecules and mucosal antigens,
resulting in antibodies with novel structural and functional mechanisms compared to circulatory
counterparts (*Flajnik, 2010*; *Kaetzel, 2014*).

Secretory (S) Immunoglobin (Ig) A is the predominant, mammalian mucosal antibody; it is a poly-
meric antibody assembled in plasma cells that link two IgA monomers and a single copy of the join-
ing chain (JC) to form dimeric (d) IgA (and to a lesser extent, higher order polymers) (*Woof and
Russell, 2011*). Following secretion in the lamina propria, dIgA is bound by the polymeric Ig recep-
tor (pIgR), a transcytotic Fc receptor (FcR) expressed on the basolateral surface of epithelial cells.
The pIgR uses five Ig-like domains (D1-D5) and a cytoplasmic tail to bind and transcytose JC-contain-
ing antibodies to the apical surface of epithelial cells; there, the pIgR ectodomain, called secretory
component (SC), is proteolytically cleaved, releasing the SC-dIgA complex into the mucosa where it
is called SIgA (*Figure 1A*; *Stadtmueller et al., 2016a*). In the mucosa, SIgA functions are thought to
be dominated by physical mechanisms including coating, cross-linking, agglutination, and enchained
growth of mucosal antigens; outcomes are diverse and, typically, not associated with inflammation
(*Woof and Russell, 2011*; *Pabst and Slack, 2020*). For example, SIgA-dependent enchained growth

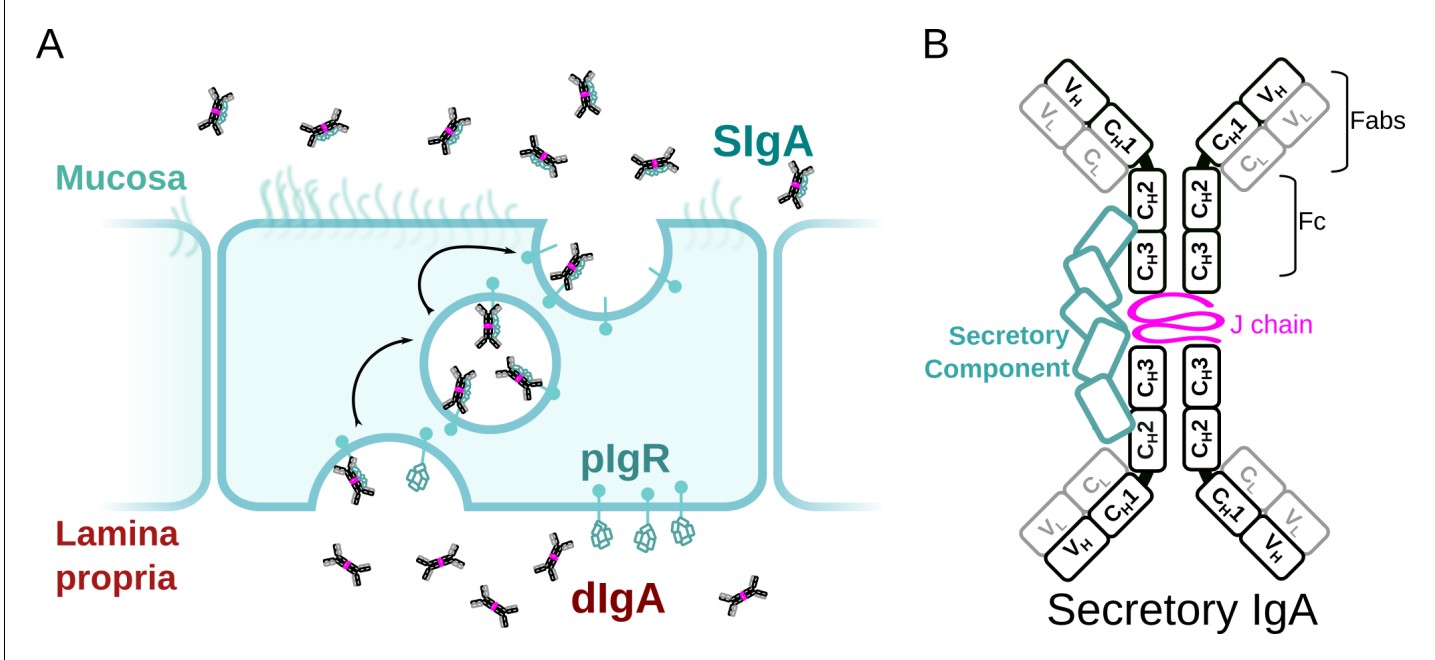

**Figure 1.** SIgA delivery to the mucosa. (**A**) Schematic depicting unliganded pIgR binding to dIgA from the lamina propria on the basolateral surface of an epithelial cell followed by transcytosis to the apical membrane and SIgA release into the mucosa. (**B**) Schematic showing protein components of SIgA, including two IgA monomers, joining chain (JC), and secretory component (SC). The IgA heavy chain is colored white with a black outline and the light chain is colored white with a gray outline. Each chain is made up of immunoglobulin domains, including IgA heavy chain constant ($C_H$1-3), heavy chain variable ($V_H$), light chain constant ($C_L$), and light chain variable ($V_L$),domains, which are indicated along with antigen-binding fragments (Fabs) and Fc regions.

The online version of this article includes the following figure supplement(s) for figure 1:

**Figure supplement 1.** Sequence identity, similarity, and alignments between mouse and human SIgA components.

of Salmonella promotes its clearance from the gut, whereas SIgA-interactions with *B. fragilis* can promote its colonization of the gut (*Moor et al., 2017*; *Donaldson et al., 2018*). Furthermore, colostrum SIgA can provide passive immunity to newborns and has been shown to have a life-long influence on microbiome composition (*Rogier et al., 2014*).

SIgA functions are supported by a unique, yet poorly understood molecular structure. Similar to other antibody classes, the IgA monomer is made up of two copies of the heavy chain and two copies of the light chain, each with variable and constant domains. Together, these chains form two antigen-binding fragments (Fabs), each containing variable domains with complementary determining regions (CDRs) that facilitate binding to a large repertoire of antigens. In contrast to monomeric IgA, SIgA is an antibody dimer and has four Fabs that are presumed to enhance binding avidity and antigen cross-linking potential when compared to antibodies with two Fabs; yet, a mechanistic understanding of how SIgA engages antigens remains a topic of investigation. IgA Fabs connect to the antibody Fc, which is made up of two copies of the IgA constant heavy chain domains, $C_H$2-$C_H$3 (*Figure 1B*). The IgA $C_H$3 contains a C-terminal extension called the tailpiece (Tp), which along with the JC, is required for IgA dimerization and subsequent pIgR binding. The Fc regions of dIgA provide binding sites for SC as well other host and microbial FcRs; however, the conformations of the dIgA Fc and pIgR domains have remained elusive along with the structures of the Tps and JC (*Stadtmueller et al., 2016a*).

To investigate SIgA structure and how it supports diverse mucosal functions, we targeted structures of SIgA from the mouse, which has been long been used as a model organism for immunological research (*Masopust et al., 2017*). Unlike humans, which have two IgA heavy chain isotypes, IgA1 and IgA2 (and two IgA2 allotypes IgA2m1 and IgA2m2) mice express a single IgA heavy chain that shares 79–80% sequence similarity and 66% sequence identity with human variants among the $C_H$2-$C_H$3 domains; the sequences of mouse SC and JC also share a high degree of sequence similarity

with human counterparts (*Figure 1—figure supplement 1*). Accordingly, we co-expressed a mouse monoclonal IgA with mouse JC and SC in transiently transfected mammalian cells, which resulted in monodisperse SIgA (see Materials and methods). Using cryoelectron microscopy (CryoEM), we determined a ~ 3.3 Å resolution structure of SIgA, which revealed a pseudosymmetric arrangement of the two IgA Fcs, bound asymmetrically to JC and SC. Comparisons with a ~ 3.3 Å resolution cryoEM structure of dIgA, revealed a dominant role for JC in maintaining the conformation of the dimer interface and geometric relationship between the two IgA Fcs. Finally, we modeled plausible positions that could be adopted by SIgA Fabs. Together, results suggest that the pseudosymmetric arrangement of SIgA core components will constrain the positions of the Fabs and influence binding of host and microbial factors.

## Results

### The structure of SIgA

We determined the CryoEM structure of mouse SIgA to a final average resolution of 3.3 Å (*Figure 2*). Local map resolution was variable, revealing side chain density for many residues, particularly at the interfaces between complex components and domains; however, some regions at the periphery of the complex were not well resolved, likely due to inherent flexibility. Local resolution was lowest for Fabs, which are mostly disordered and were not built. Additionally, sidechains were not well-resolved for the majority of residues in SC D2 and some residues in the $C_H2$ domains (*Figure 2—figure supplement 1*, *Figure 2—figure supplement 2*, *Figure 2—figure supplement 3*). The refined structure revealed a pseudosymmetric assembly of two IgA monomers conjoined at the center by the JC and bound by SC. The $C_H2$-$C_H3$ domains in both Fcs aligned with the published monomeric IgA Fc structure lacking the Tp (*Herr et al., 2003*) (not shown); however, despite shared sequences, the four IgA heavy chains formed structurally unique contacts with the JC and SC. To distinguish these differences, we designated a unique ID for each heavy chain: A, B, C, or D, which we also use to describe corresponding $C_H$ domains and Fcs (*Figure 2*). The $Fc_{AB}$ and $Fc_{CD}$ are bent and tilted with respect to each other, resulting in distinct concave and convex surfaces on the complex. To describe the conformation, we defined the angle between the centroid axes of $Fc_{AB}$ and $Fc_{CD}$ (97 degrees) as 'bend', and the angle between the two non-intersecting centroid planes of $Fc_{AB}$ and $Fc_{CD}$ (30 degrees) as 'tilt'. SC asymmetrically contacts the same face of both $Fc_{AB}$ and $Fc_{CD}$ along what we define as the 'front face' of the molecule. Potential N-linked glycosylation sites (PNGS) are distributed throughout the complex; the seven, PNGS located on SC, are clustered on the front face of SIgA (*Figure 2*). The map revealed well-ordered carbohydrates at a subset of these sites, which were modeled in the structure (*Figure 2—figure supplement 4*).

### The structure of the Fc-JC interface

The SIgA structure reveals numerous molecular interactions that effectively lock the Fc regions from two IgA monomers with a single JC to form a centrally located interface. The interface is dominated by the four heavy chain Tps ($Tp_A$, $Tp_B$, $Tp_C$, $Tp_D$) and the first half of the JC sequence, or, 'core' ($JC_{core}$), which together fold into a single β-sandwich-like domain at the center of the complex. The interface is further stabilized by the second half of the JC sequence, which folds into two beta-hairpin 'wings' ($JC_{W1}$ and $JC_{W2}$), each extending outward from the center of the complex and binding one $C_H3$ from each IgA monomer (*Figure 3*).

Notably, whereas the four Tps share identical sequence, they form unique interactions with each other and the JC. $Tp_C$ and $Tp_D$ and the first three JC β-strands form one side of the β-sandwich, oriented toward the front face of SIgA, while $Tp_A$, $Tp_B$ and the subsequent β-strand from the JC form the other side, oriented toward the back face of SIgA (*Figure 3A,B*). Despite their location on opposing sides of the complex, $Tp_A$ and $Tp_C$ adopt similar conformations that are stabilized though β-sheet interactions with $Tp_B$ and $Tp_D$, respectively, as well as interactions between C-terminal residues and two unique pockets formed in part by the $JC_{core}$ (*Figure 3C*). The cornerstones of these interactions are $Tp_A$ and $Tp_C$ penultimate cysteine residues (Cys 466), which disulfide bond to $JC_{core}$ Cys68 and $JC_{core}$ Cys14, respectively, and thereby lock both IgA monomers to the JC. Additional stability is provided by $Tp_A$ and $Tp_C$ ultimate tyrosine residues (Try467), which in addition to binding $JC_{core}$ residues form contacts with $C_H3_C$, and SC D1, respectively (*Figure 3C*). $Tp_B$ and $Tp_D$ form β-

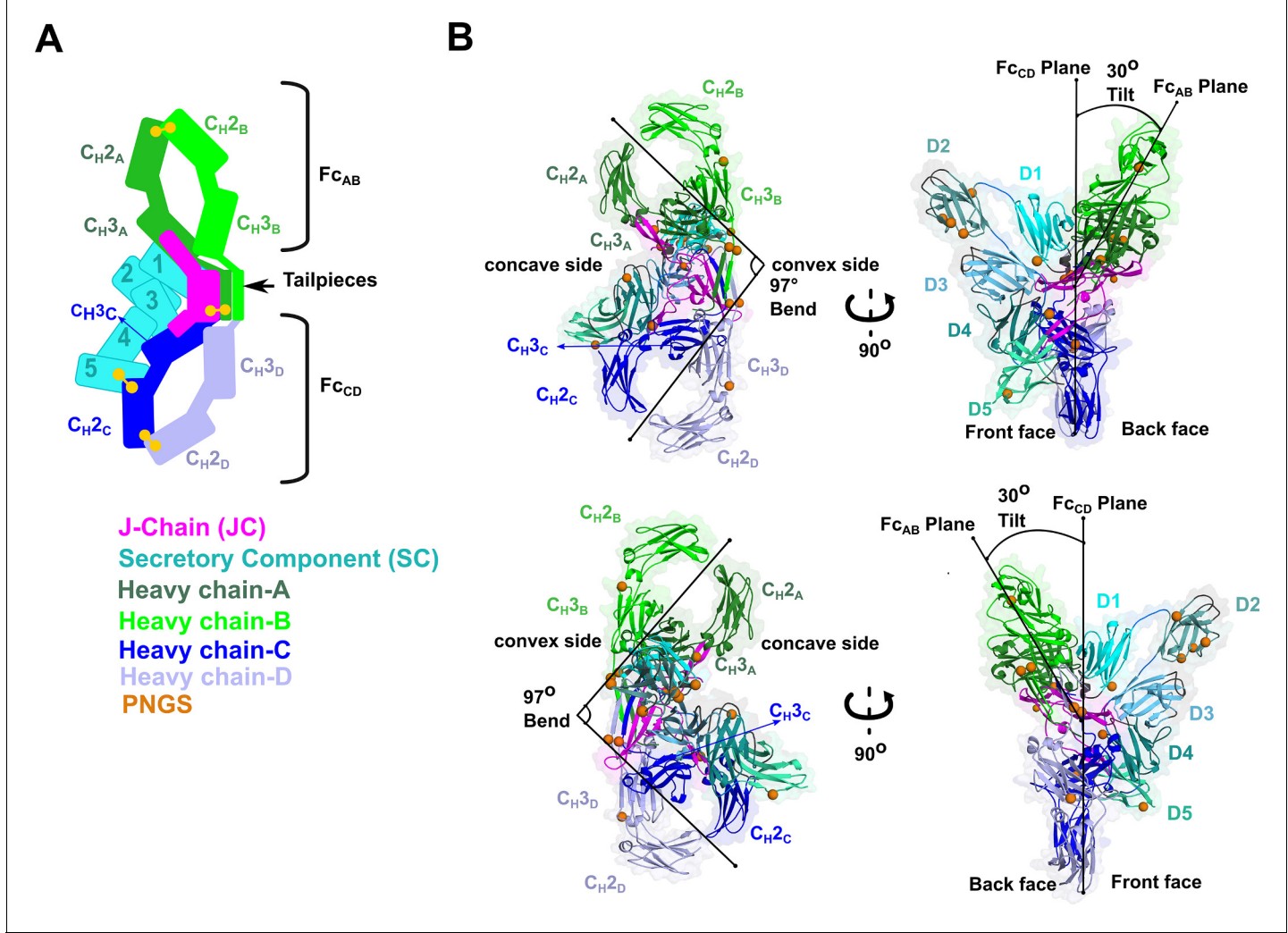

**Figure 2.** SIgA structure. (**A**) Schematic representing ordered domains in the SIgA structure. The relative position of each SIgA component is approximated based on the structure. Chain IDs and corresponding $C_H$ domains and Fcs are labeled along with SC domains (1-5). Each SIgA component is depicted in a unique color; J chain (JC), magenta; secretory component (SC), teal; heavy chains A, dark green, B, light green, C, dark blue, and D, light blue, respectively. (**B**) Cartoon representation (with semi-transparent molecular surface) of the SIgA structure shown in four orientations and colored as in (A). $C_H$ and SC domains (D1-D5) are labeled and PNGS are shown as orange spheres. The bend and tilt between the two Fcs are indicated with a line and the angle measured in the structure; the concave and convex sides are labeled, along with the front face and the back face.

The online version of this article includes the following figure supplement(s) for figure 2:

**Figure supplement 1.** SIgA cryoEM data collection and CryoSparc processing pipeline.
**Figure supplement 2.** SIgA cryoEM data collection and Relion processing pipeline.
**Figure supplement 3.** SIgA cross-validation FSC curves.
**Figure supplement 4.** SIgA glycosylation.

sheet interactions with $Tp_A$ and $Tp_C$, respectively, but in contrast to $Tp_A$ and $Tp_C$, form limited interactions between their C-terminal residues and other complex components. For example, although $Tp_D$ Cys466 is visible in the cryoEM map, we fail to find evidence that $Tp_B$ or $Tp_D$ Cys466 are involved in disulfide bonds and the seven C-terminal residues of $Tp_B$ are disordered. Notably, however, $Tp_D$ Tyr467 is positioned to contact two arginine residues in the $JC_{core}$ as well as conserved residues in SC D1, signifying another type of conserved interaction for the Tp ultimate tyrosine residues that complements those observed for $Tp_A$ and $Tp_C$ Try467 (**Figure 3C**).

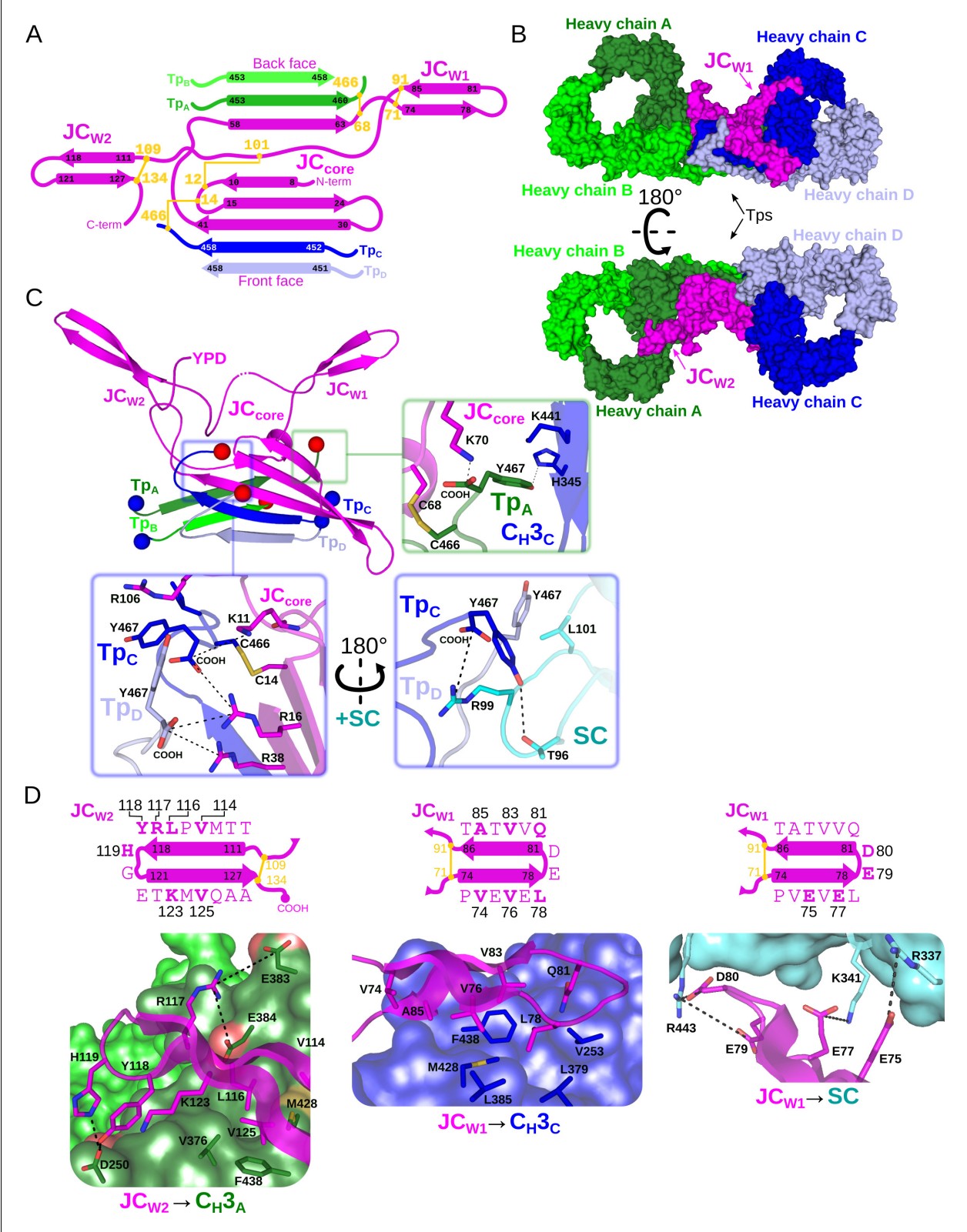

**Figure 3.** Fc dimer interface and JC structure. (**A**) Schematic depicting the topology that arises from JC and its interactions with four Tps and colored as in *Figure 2*. JC regions (JC$_{W1}$, JC$_{W2}$, JC$_{core}$, and N- and C termini) are labeled along with Tp$_{A-D}$ and their location relative to the front face or back face of SIgA; disulfide bonds are indicated in yellow and the residue boundaries of each β-strand are labeled. (**B**) SIgA molecular surface representation (SC removed) colored as in *Figure 2* and indicating the location of JC$_{W1}$, JC$_{W2}$ and Tps relative to each HC$_{A-D}$ in two SIgA orientations. (**C**) Cartoon

*Figure 3 continued on next page*

*Figure 3 continued*

representation of JC and Tps complex (same region as shown in panel A) with the N- and C termini of each Tp shown as blue and red spheres, respectively. Regions surrounding the three C-terminal residues of Tp are boxed, $Tp_A$ (green box), and $Tp_C$ and $Tp_D$ (blue boxes), and enlarged. Enlargements show Tp carboxy termini (COOH) and side chain sticks involved in interactions with adjacent Tps, JC, $C_H3$ domains, and SC. The three C-terminal residues of $Tp_B$ are disordered and not shown. (**D**) Topology diagram and sequence (top) and structure (bottom) detailing interactions between $JC_{W2}$ and $C_H3_A$ (left), $JC_{W1}$ and $C_H3_C$ (center), and $JC_{W1}$ and SC (right) and colored as in A-C. JC residues interacting with $C_H3$ or SC are indicated in bold and numbered in the topology diagram (top) and shown as sticks on a cartoon representation (bottom); $C_H3$ and SC domains are shown as molecular surface representations with interfacing residues shown as sticks. Negatively charged atoms in $C_H3$ that form contacts are shown as a red surface.

The two JC wings, $JC_{W1}$ and $JC_{W2}$, represent a second set of unique interactions that help to lock the two IgA Fc monomers together. $JC_{W1}$ and $JC_{W2}$ are unique in sequence yet both form anti-parallel β-sheets, enclosed on the N- and C-terminal ends by intra-JC disulfide bonds. $JC_{W1}$ is composed primarily of hydrophobic and acidic residues and uses hydrophobic interactions to bind the front face of $C_H3_C$, whereas acidic residues in $JC_{W1}$ mediate interactions with SC (*Figure 3D*). $JC_{W2}$ is five residues longer than $JC_{W1}$ and shares a larger, more chemically diverse interface with the back face of $C_H3_A$, involving several electrostatic interactions (*Figure 3D*). Notably, although residues forming the interface between each JC wing and its respective $C_H3$ domain differ, both wings bind a similar location on $C_H3_A$ and $C_H3_B$, illustrating a mechanism by which a single JC can bind two identical antibodies uniquely.

## Secretory component

SC, the pIgR ectodomain, has five domains, D1-D5, each having an Ig-like fold with loops structurally similar to antibody CDRs. The crystal structure of unliganded human SC demonstrated that these domains adopt a compact arrangement, in which a subset of residues in three D1 CDR-like loops (hereafter CDR) interface with residues in D4 and D5 to form a closed conformation (*Figure 4A*; *Stadtmueller et al., 2016a*). In contrast to unliganded SC, the SIgA structure reveals SC domains bound to dIgA in an elongated, open conformation contacting both antibodies and the JC asymmetrically on the front face of the complex (*Figure 4A*). D1 contacts $Fc_{AB}$, $Fc_{CD}$ and the JC, D2 protrudes away from other SC domains and the front face of the molecule, and D3 is clamped to D1, positioning D4 and D5 to contact $Fc_{CD}$ and $JC_{W1}$ (*Figure 4A*).

Interactions between SC D1 and dIgA components are of special interest because residues in D1 CDR loops are reportedly necessary for binding to JC-containing antibodies, yet are also involved in stabilizing SC in its unliganded conformation (*Brandtzaeg, 2013*; *Stadtmueller et al., 2016a*). Indeed, the SIgA structure reveals residues in or adjacent to CDR1, CDR2, and CDR3 bridging interactions with JC, $C_H3_A$, $C_H3_B$, $Tp_C$, and $Tp_D$, rather than binding to SC D4 or D5 as observed in the closed conformation (*Figure 4A,B*). The D1 CDR1 appears to play a key role binding JC, primarily through a conserved Try-Pro-Asp motif located in the JC C-terminal loop following $JC_{W2}$. This interaction is dominated by D1 CDR1 Arg31 and His32, which hydrogen bond to JC Asp137 thereby linking D1 to the C-terminus of JC. An additional contact, between D1 Asn30 and JC 106Arg, appears to link D1 to the loop connecting $JC_{W1}$ and $JC_{W2}$ (*Figure 4B*). In turn, residues flanking CDR1, and residues in and adjacent to CDR2, mediate contacts with two, non-overlapping sites on $C_H3_A$ and $C_H3_B$ (*Figure 4B*). In contrast to CDR2, which binds the $Fc_{AB}$, D1 CDR3 residues bind the $Fc_{CD}$, contacting the C-terminal regions of $Tp_C$ and $Tp_D$. Notably, CDR3 Leu101 contacts the ultimate residue, Tyr467, in both $Tp_C$ and $Tp_D$, highlighting the dual role that the conserved ultimate tyrosine residues in the Tps play in binding to JC and SC residues (*Figures 3C* and *4B*).

Compared to D1, D2-D5 form limited interactions with dIgA components; yet, we find each engaged in a unique position that stabilizes the SIgA structure. D2 does not contact dIgA directly; instead, the D1-D2 linker is extended, positioning D2 at the outer edge of the front face where it shares a minimal, ~250 $Å^2$, interface with D3. The map surrounding D2 is poorly ordered suggesting that the position of D2 is flexible to the extent allowed by the linkers between D1-D2 and D2-D3. Together with D2, D3-D4-D5 adopt a near-linear arrangement, or arm, that is bent approximately 48 degrees relative to D1, positioning D4-D5 to contact the $Fc_{CD}$ front face and the $JC_{W1}$ (*Figure 4A, B*). This conformation is stabilized by a 324 $Å^2$ interface between D1 and D3, involving conserved hydrophobic residues from both domains (*Figure 4C*). The interface appears to function as a

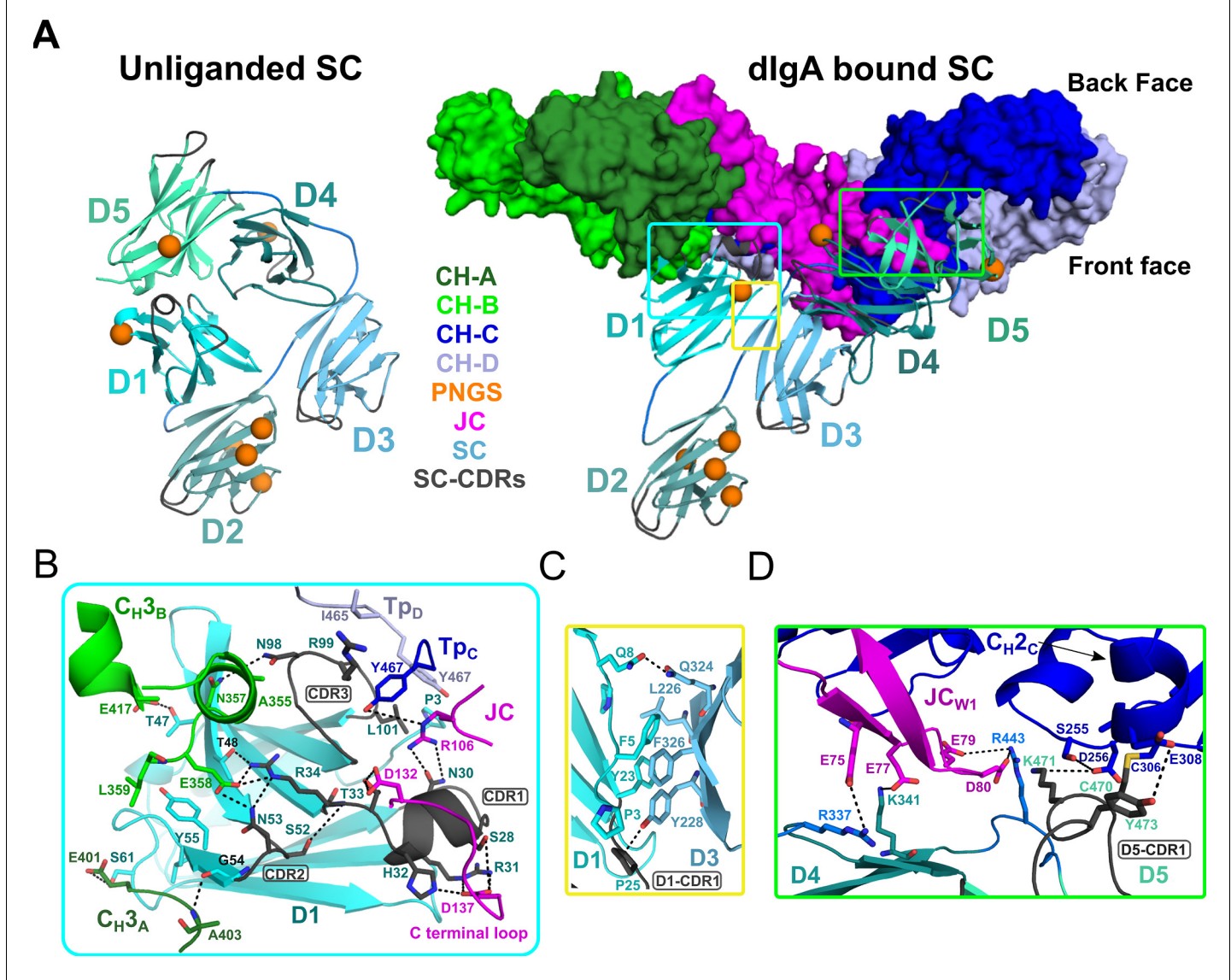

**Figure 4.** SC Structure. (A) Cartoon representations of an unliganded mouse SC homology model based on unliganded human SC crystal structure (PDB code 5D4K; left) and the dIgA-bound mouse SC (right). The dIgA is shown as a molecular surface representation; SC domains D1-D5 are show in color gradient from cyan (D1) to pale green (D5), with CDR loops colored gray; PNGS are shown as orange spheres. Regions where SC interfaces with dIgA and other SC domains are boxed. (B-D) Boxed, enlarged views of stabilizing interfaces between SC Fc, JC and other SC domains; colored as in panel A. (B) SC D1 interface with $C_H3_A$, $C_H3_B$, $Tp_C$, $Tp_D$. (C) SC D1-D3 interface. (D) SC D4-D5 interface with JC and $C_H2_C$. In all panels conserved, interfacing residues are shown as sticks and hydrogen bonds and salt bridges are show as black dashes.

keystone in a D1-D3-D4-D5 bridge connecting $Fc_{AB}$ and $Fc_{CD}$ and contrasts with SC's unliganded structure, in which D1 and D3 do not share an interface (*Figure 4A*). Although D3 does not contact either of the Fcs or the JC, its position is further stabilized by a 646 Å$^2$ interface with D4 (distinct from the D3-D4 interface in the closed conformation) and several D3-D4 linker residues that are positioned to contact $JC_{W1}$ (*Figures 3D* and *4D*). D4 and D5 share a similar interface in both unliganded SC and SIgA; however, the domains are repositioned relative to other domains such that residues in D4 and D5 that contact D1 in SC's unliganded conformation, are solvent exposed. Together, D4-D5 engage the front face of $Fc_{CD}$ through a patch of residues that surround $C_H2_C$ Cys306 and the tip of $JC_{W1}$. The interface is dominated by residues in D5 CDR1, including Cys470, which disulfide bonds to $C_H2_C$ Cys306, covalently linking SC to the antibody, as well as D5 CDR1 Lys471 and Tyr473, which form hydrogen bonding and hydrophobic interactions with $C_H2_C$

(*Figure 4D*). Additional stability at this interface is provided by a handful basic residues located in D5 and linkers connecting D3-D4 and D4-D5, which form electrostatic interactions with acidic residues in $JC_{W1}$ and effectively sandwich $JC_{w1}$ in-between $C_H2_C$ and SC (*Figures 3D* and *4D*).

## Structure of dIgA

The SIgA structure revealed a bent and tilted relationship between two IgA monomers, which was stabilized in part by a bridge of interactions between SC domains. To investigate SC's contribution to the conformation of SIgA, we determined a 3.3 Å average resolution cryoEM structure of dIgA, which contained the same components as SIgA, except for SC (*Figure 5*, *Figure 5—figure supplement 1*, *Figure 5—figure supplement 2*). The dIgA structure revealed two IgA monomers arranged similar to those in SIgA; Fabs were disordered. The centroid axes of $Fc_{AB}$ and $Fc_{CD}$ were

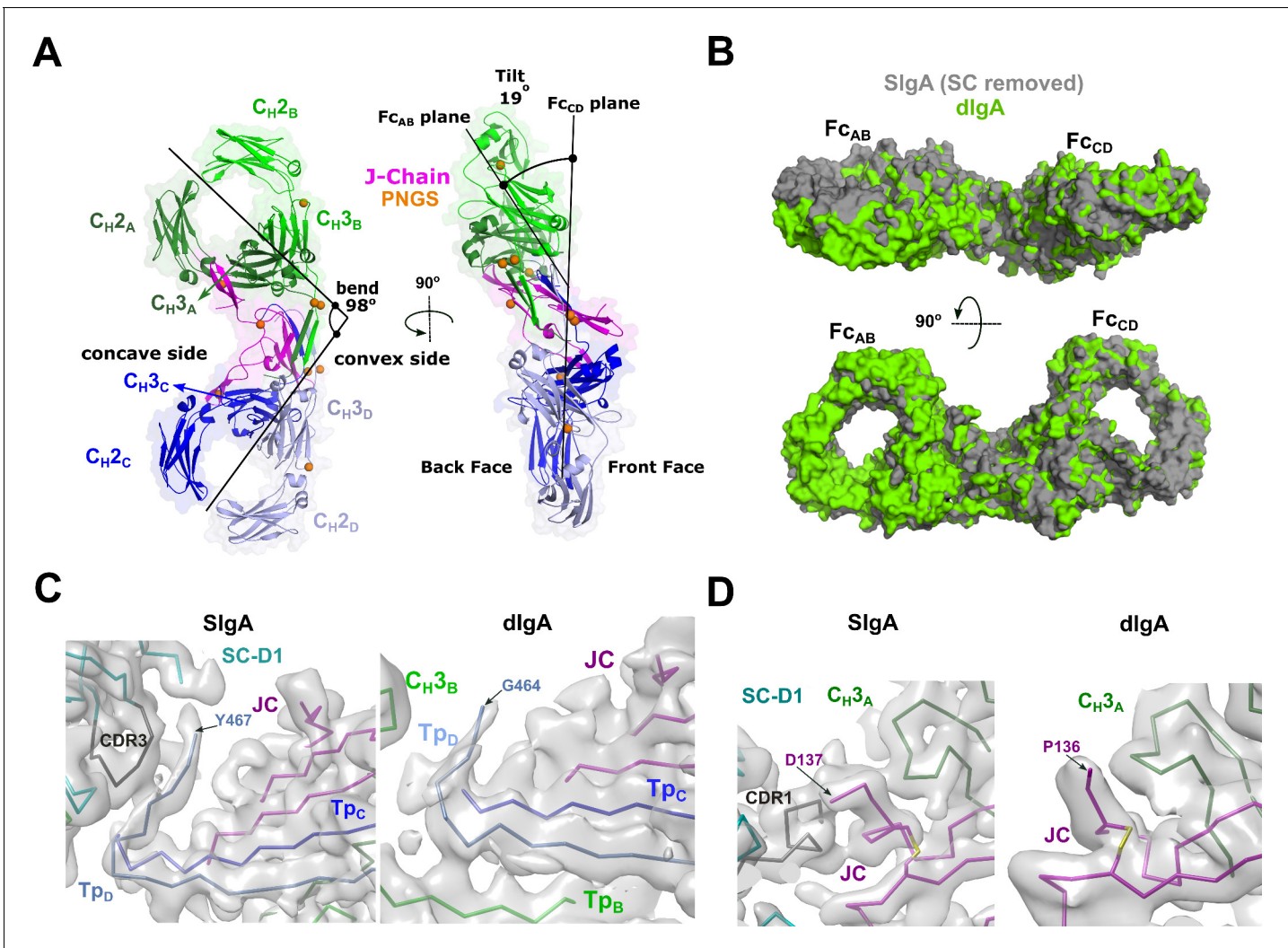

**Figure 5.** dIgA structure. (A) Cartoon representation (with semi-transparent molecular surface) of the unliganded dIgA structure shown in two orientations and colored as in *Figure 2*. The bend and tilt between the two Fcs is indicated with a line and the angle measured in the structure; the concave and convex sides are labeled, along with the front face and the back face and $C_H$ domains. PNGS are shown as orange spheres. (B) Molecular surface representation of SIgA (gray; SC removed) aligned to dIgA (green) on the JC from each complex structure. (C) CryoEM maps and structures for SIgA (right) and dIgA (left) detailing the region surrounding $Tp_D$. (D) CryoEM maps and structures for SIgA (right) and dIgA (left) detailing the region surrounding the JC C-terminus.

The online version of this article includes the following figure supplement(s) for figure 5:

**Figure supplement 1.** dIgA cryoEM data collection and processing strategy.

**Figure supplement 2.** dIgA cross-validation FSC curves.

bent 98 degrees relative to each other (compared to 97 degrees in SIgA), and the centroid planes were tilted at an angle of 19 degrees (compared to 30 degrees in SIgA) (*Figure 5A*). These results suggest that the bend in SIgA is conferred by JC binding to heavy chains and/or interactions between the Tps from both antibodies. The 11-degree difference in tilt between the two structures suggests that SC binding can influence the conformational relationship between the two IgAs. Aligned individually, the structures of the heavy chains and JC were largely superimposable with equivalent chains in SIgA; however, differences were apparent in regions that are bound by SC in SIgA (*Figure 5B*). In particular, Tp$_D$, which forms part of the SC D1-JC interface in SIgA, adopts a different conformation, and the three C-terminal residues are disordered (*Figure 5C*). The three C-terminal residues of JC (Tyr-Pro-Asp), which bind SC D1 CDR1 in SIgA, are also disordered in dIgA (*Figure 5D*).

## SIgA structure impact on antigen binding

The SIgA and dIgA structures revealed bent and tilted relationships between two IgA monomers, which are supported by stabilizing interactions that are likely to limit flexibility between the two IgA Fcs. Seeking to better visualize how this geometry could influence the flexible positions of SIgA Fabs, we modeled potential Fab locations onto SIgA structures using a computational conformational-search approach, which approximated the position of each Fab by mapping a vector (beginning at N-terminus of C$_H$2 and ending at the center of mass of Fab CDRs) onto a Fibonacci spherical lattice (FSL)(*Marques et al., 2013*; *Figure 6—figure supplement 1*). We reasoned that this approach would broadly survey possible Fab positions without the constraints of the diverse Fc-Fab linkers and contacts between heavy chains and light chains found in SIgA among mammals (*Woof and Kerr, 2006*). This strategy identified 8000 possible conformations for each Fab, from which those that clashed with Fc, JC, or SC were eliminated. Inspecting all SIgA models revealed a distribution, in which Fab CDRs dominated positions on the concave side of SIgA. Notably, this distribution revealed a large exposed surface area on the convex side of SIgA (C$_H$2$_B$-C$_H$3$_B$ and C$_H$2$_D$-C$_H$3$_D$) that was never occupied by the Fabs (*Figure 6A*). To quantify distributions for each of the four Fab's CDRs, we measured the angle of each vector relative to the center of mass of the JC and Fcs along the Fc$_{AB}$ plane (*Figure 6B*). The distribution revealed a dominance of angles less than 90 degrees, consistent with Fabs occupying more positions on the concave side of SIgA, but also a unique distribution for each Fab, collectively illustrating how the complex pseudosymmetry (bend and tilt relative to the Fc$_{AB}$) might influence the positions of CDRs and associated interactions with antigen.

## Discussion

Our work describes the structures of mouse SIgA and its precursor, dIgA, which lacks the SC. Together, these structures provide a glimpse of one of the most abundant and long-studied mammalian antibodies. Among other discoveries, decades of prior work determined the structures of unliganded SC and monomeric IgA Fc lacking the Tps, provided biochemical insights on SIgA components, and also revealed many key elements of pIgR-dependent transcytosis, mucosal antibody evolution and function. (*Herr et al., 2003*; *Bonner et al., 2009*; *Woof and Russell, 2011*; *Brandtzaeg, 2013*; *Stadtmueller et al., 2016a*). Yet without the structures of the JC, dIgA and SIgA, interpreting biological data, understanding functional mechanisms and designing IgA-based therapeutics has remained a challenge. Our structures and related, human IgA structures published in the same timeframe as this work (*Kumar et al., 2020*; *Wang et al., 2020*) provide an opportunity to addresses many outstanding questions surrounding SIgA and its functions as a mucosal antibody; we discuss a subset of them below and provide a model for formation, transport, and function of SIgA in mammals.

## The role of JC in dimeric IgA assembly

The JC plays an essential and early-stage role in SIgA production by linking two or more IgA monomers and conferring the ability of the resulting antibody polymer (e.g. dIgA) to bind the pIgR ectodomain (*Woof and Russell, 2011*). Yet, its evolutionary origin, its structure (even its fold), and how it facilitates antibody polymerization remained elusive through decades of research. Our dIgA and SIgA structures resolve some of these uncertainties by revealing the JC fold, how it is integrated

with the Tps from four IgA heavy chains, and how it is stabilized by contacts between the JC wings and IgA $C_H3$ domains. The structure does not appear to be closely related to proteins or complexes with other functions because our searches of the protein data bank have failed to identify structures with high similarity to the JC or the JC-Tp assembly; although, we cannot rule out the possibility that the unliganded JC might adopt a different conformation with similarity to determined structures. For now, this leaves the JC relationship to other proteins, and structural clues on the JC evolutionary origin, uncertain.

The molecular contacts observed in the dIgA and SIgA structures reveal, in part, how the JC facilitates antibody polymerization. Central to this mechanism appears to be the JC's ability to bind each of the four identical heavy chain sequences uniquely, resulting in pseudosymmetric, dimeric antibody. This mechanism may represent an efficient way for the JC to link two antibodies; yet, it also appears likely to influence the function of the final product, SIgA, because our data indicate that JC plays a dominant role in inducing and/or maintaining the bent relationship between the two Fcs, which our modeling suggests can influence Fab positions. Despite these observations, many details of JC-dependent antibody polymerization remain outstanding; in particular, the order of events during which the JC binds two IgA monomers to assembly dIgA is unclear. The process appears likely to involve conformation changes in both the JC and the IgA monomers because in the absence of JC, monomeric IgA (the form typically found in serum) is functionally active (*Woof and Russell, 2011*), suggesting that the interactions we observe between Tps from two IgA monomers do not form, or are not stable, without the JC. Similarly, in the absence of contacts with IgA, the conformation of JC we observe in SIgA and dIgA is not likely to be stable, suggesting that the JC's unliganded structure is unknown. These observations point to a polymerization process that involves conformational changes in the JC and two IgA heavy chains, which promote β-sheet like interactions between Tps and/or Tps and the JC as well as disulfide bond formation between chains. Recent reports also suggest that molecular chaperons are involved in dIgA assembly (*Suzuki et al., 2019*; *Xiong et al., 2019*).

## pIgR binding to dIgA

Following assembly, dIgA binding to the pIgR ectodomain, or SC, is required for SIgA delivery to the mucosa. When unliganded, SC domains adopt a compact conformation, in which conserved residues in D1 form an interface with D4 and/or D5 (*Stadtmueller et al., 2016a*). The SIgA structure reveals a marked rearrangement of SC domains that positions residues in the D1 CDRs in direct contact with $C_H3_A$, $C_H3_B$, $Tp_C$, $Tp_D$ and JC and positions D5 CDRs in contact with $C_H3_C$. This conformational change is consistent with published double electron-electron resonance (DEER) distance measurements showing a 70 Å separation between nitroxide spin labels attached to SC D1- residue 67 and D5 residue 491 in human SIgA (*Stadtmueller et al., 2016a*); the distance between Cα atoms of equivalent residues in the mouse SIgA structure, D1 Ile67 and D5 His493, is 61 Å. The SC crystal structure and DEER experiments led to a model, in which accessible motifs in D1 contact dIgA in a recognition binding event that triggers a conformational change allowing previously buried D1 motifs and accessible D5 motifs to bind dIgA (*Stadtmueller et al., 2016a*). Our structure is consistent with this model, revealing that conserved D1 residues Arg31, His32, Arg34, Thr48 and Tyr 55, which bind D4 or D5 in unliganded SC, mediate interactions with $C_H3_B$ and JC. It is yet unclear which residues mediate the recognition binding event; however, one possibility is that SC D1-CDR3 residues, which are exposed in unliganded SC, form the initial contacts with the tailpieces ($Tp_C$ and $Tp_D$) and $C_H3_B$, which are exposed in dIgA (*Figure 5C*). Our structures suggest that during the conformational change, D2 moves away from D1 while D3 forms hydrophobic interactions with D1 in a process that positions the D4-D5 arm near the $FC_{CD}$ and D5 in contact with $C_H2_C$.

The pIgR (and SC) has long been known to bind only JC-containing molecules (*Woof and Russell, 2011*), a requirement that is validated by the SC-JC interfaces in the SIgA structure. However, it also appears likely that the JC indirectly supports SC binding by inducing the bend that positions $C_H3_B$ (which binds D1) and $C_H2_C$ (which binds D5) optimally to be bound by the SC bridge comprising D1-D3-D4 and D5. D2 is absent from this bridge, and does not appear to contact the heavy chains or JC, suggesting that despite its reported contribution to dIgA-binding kinetics (*Stadtmueller et al., 2016a*), its role is indirect. The distal location of D2 in the SIgA structure is also consistent with a model in which SC from birds, reptiles and amphibians, which are lacking the D2 domain (*Stadtmueller et al., 2016b*), would bind analogous to D1-D3-D4-D5 in our SIgA structure.

## The functional significance of SC

Although the pIgR plays a critical role in delivering SIgA to mucosal secretions, functionally, why its ectodomain (SC) remains attached to SIgA is less clear. While SC may play a role in stabilizing the conformation of SIgA, the dIgA structure suggests that SC binding induces relatively small changes in the bent and tilted relationship between the two IgAs. SC has been reported to protect SIgA from degradation and to play a role in binding to host and bacterial factors (*Kaetzel, 2005*). Indeed, our structure reveals SC forming extensive interfaces with the heavy chains and the JC where it may protect areas especially vulnerable to proteolysis; however, roughly 16% of the SIgA core (not including Fabs) is occluded by SC, leaving the majority of the molecule exposed. Being located on one face of the molecule, SC exhibits significant accessible surface area (in excess of 25,000 $\text{Å}^2$) leaving it well-positioned to interact with host or microbial factors. D2 is particularly accessible, being located distal from SIgA's center and exhibiting evidence of flexibility. It also includes four of the seven PNGS which could facilitate carbohydrate-mediated binding events, although the number of PNGS on each SC domain is variable among species. It is unclear why mammalian SC evolved to include D2; our observations point toward a role in mediating interactions with host and/or microbial factors (*Figure 7*).

## Comparisons with human SIgA

SIgA is found in all mammals; however, some species including humans, express multiple isoforms, and allotypes of the IgA heavy chain, which are associated with unique expression patterns and functions in vivo (*Woof and Russell, 2011*). Furthermore, differences in IgA heavy chain $C_H2$ and $C_H3$, JC and SC sequences have also been associated with variable glycosylation patterns, variable interactions with FcRs and differences in potential to form higher order polymers (e.g. tetrameric SIgA) (*Woof and Russell, 2011*), suggesting that on some level, SIgA structures are heterogeneous. Understanding SIgA structural heterogeneity is important for understanding function, especially between mouse and human IgAs given the long history of using mouse model systems for immunological research (*Masopust et al., 2017*). Our structures include the $C_H2$ and $C_H3$ domains from the single IgA heavy chain found in the mouse genome, which along with JC and SC sequences share 65–77% identity and 79–86% similarity with human counterpart sequences (*Figure 1—figure supplement 1A*). Analysis of our structures and comparison of human and mouse SIgA component sequences revealed a handful of differences at interfaces between complex components, including contacts between the JC wings and SC or $C_H3$ (*Figure 1—figure supplement 1B*). However, the majority of interfacing residues appear to be conserved (*Figure 1—figure supplement 1B*). Consistent with these observations, comparison of our mouse SIgA with structures of dimeric forms of the human SIgA1 core (no Fabs), reported in the same timeframe as this work (*Kumar et al., 2020*; *Wang et al., 2020*) revealed largely superimposable structures with some variability in contacts between complex components (not shown). The modeled positions of SC D2 are variable among structures, likely due to inherent flexibility in the domain's position, however, the structures have similar angles of bend (99 degrees in human SIgA and 97 degrees in mouse SIgA) and tilt (31 degrees in human SIgA and to 30 degrees tilt in mouse SIgA). Although additional studies will be needed to determine if and how sequence variability in SIgA components might uniquely influence SIgA assembly and functions, the similarities we observe suggest that the core Fc region of SIgA variants found in mammalian mucosa adopt largely conserved structures characterized by limited conformational flexibility, which our modeling suggests will constrain the positions that Fabs can occupy.

## SIgA functions

Our data provide a broad range of structural findings, addressing long-outstanding questions regarding the structure of the IgA Tps, the JC and SC, yet how the structure supports SIgA's divergent roles in pathogen clearance and microbial homeostasis remains an open topic of investigation. In the mucosa, SIgA encounters a broad range of binding partners and antigens. Besides pIgR, a number of FcR are known to interact with IgA, including pathogen receptors such as SSL7, Arp4 or Sir22, and host receptors such as FcαR1, although FcαR1 is absent from the mouse genome. IgA FcRs typically bind a canonical site located near $C_H2$-$C_H3$ elbow (*Herr et al., 2003*; *Ramsland et al., 2007*; *Kazeeva and Shevelev, 2009*). In the SIgA and dIgA structures, two of the four accessible

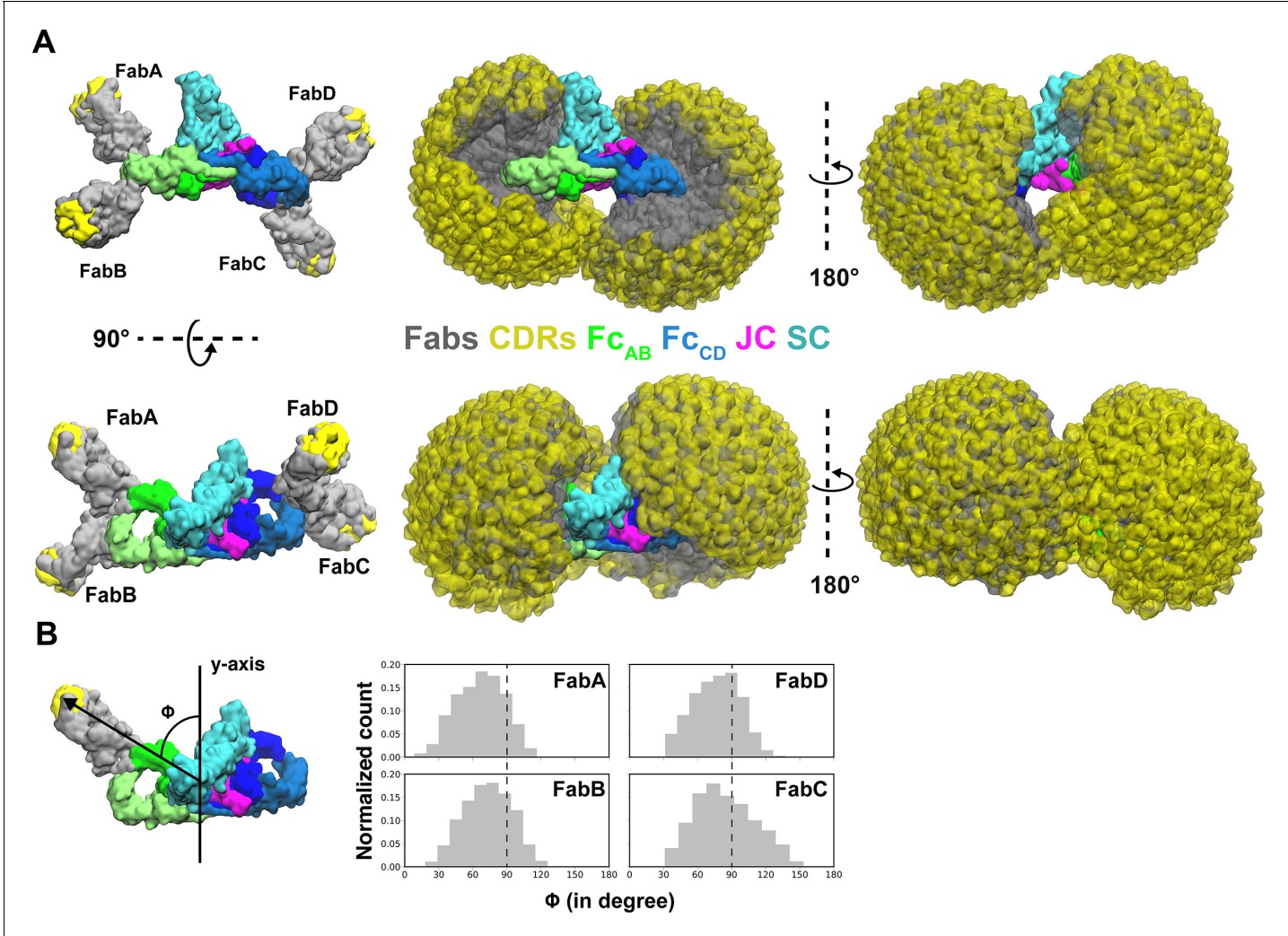

**Figure 6.** Modeling of Fabs and CDRs on SIgA. (**A**) SIgA structure shown with four Fabs, each modeled in a single position as well as all possible positions and shown in multiple orientations. Complex components are colored according to the key (center). (**B**) The number of positions sampled by Fab CDRs were quantified by measuring the angle (φ) between the Fab vector and the y-axis, a vector parallel to the FcAB plane and passing through the center of mass of Fc and JC. The frequency (normalized count) of each angle is shown as a histogram.

The online version of this article includes the following figure supplement(s) for figure 6:

**Figure supplement 1.** Modeling strategy for the computational search of SIgA Fab positions.

sites are occluded by JC wing interactions with $C_H3_A$ and $C_H3_C$. This leaves two sites, on $C_H3_B$ and $C_H3_D$, accessible for FcR binding. Notably, these sites are located on the convex edge of the complex, which our computational modeling predicts, is never occluded by Fabs (*Figures 6* and *7*). This arrangement would provide an unobstructed approach for FcRs to bind SIgA, which in the context of host cell receptors would promote favorable outcomes. On the other hand, it could leave SIgA especially vulnerable to pathogen FcRs, the binding of which might lead to Fab cleavage or other outcomes that would be detrimental to the host. We also anticipate that the bent and tilted relationship between the two IgAs, and predicted directionality of the two sets of Fabs, would influence how SIgA engages antigen. In the context of endogenous mammalian antibodies, unique properties of different $C_H2$ -$C_H1$ linkers and stabilizing interactions between the heavy chains and light chains are likely to further dictate Fab positions; however, perhaps both sets of Fabs being directed toward the concave side would stabilize binding to flexible antigens, to carbohydrates, or perhaps influence the strength of antigen crosslinking (*Figure 7*).

# Materials and methods

**Key resources table**

| Reagent type (species) or resource | Designation | Source or reference | Identifiers | Additional information |
|---|---|---|---|---|
| Other | IGHA_MOUSE | UniProtKB RRID:SCR_004426 | UniProtKB: P01878 | Amino acid sequence of *Mus musculus* IgA heavy chain constant regions |
| Other | LAC1_MOUSE | UniProtKB RRID:SCR_004426 | UniProtKB: P01843 | Amino acid sequence of *Mus musculus* lambda-1 light chain constant region |
| Other | IGJ_MOUSE | UniProtKB RRID:SCR_004426 | UniProtKB: P01592 | Amino acid sequence of *Mus musculus* joining chain |
| Other | PIGR_MOUSE | UniProtKB RRID:SCR_004426 | UniProtKB: O70570 | Amino acid sequence of *Mus musculus* polymeric Ig Receptor |
| Other | STA121 | *Moor et al., 2017* | | Variable region of human IgA2 antibody STA121 |
| Cell line (*Homo sapiens*) | HEK Expi293F | Thermo Fisher RRID:SCR_008452 | Cat#: A14635 RRID:CVCL_D615 | |
| Recombinant DNA reagent | pD2610-v1 | ATUM | Cat#: pD2610-v1-03 | Mammalian expression plasmid |
| Transfected construct (*Homo sapiens*) | STA121 IgA Heavy chain (HC) in pD2610v1 | Materials and methods of this paper | | Construct encoding STA121 heavy chain (HC) variable region fused to *Mus musculus* IgA HC constant domains |
| Transfected construct (*Homo sapiens*) | STA121 IgA light chain (LC) lambda in pD2610v1 | Materials and methods of this paper | | Construct encoding STA121 light chain (LC) variable region fused to *Mus musculus* lambda LC constant domain |
| Transfected construct (*Homo sapiens*) | Mouse Secretory component (SC) in pD2610v1 | Materials and methods of this paper | | Construct encoding residues 1–567 of *Mus musculus* pIgR (a.k.a. secretory component; SC) |
| Transfected construct (*Homo sapiens*) | Mouse joining chain (JC) in pD2610v1 | Materials and methods of this paper | | Construct encoding *Mus musculus* joining chain (JC) |
| Peptide, recombinant protein | STA121 Secretory IgA | Materials and methods of this paper | | Protein complex produced from transfected constructs and containing: STA121 HC, STA121 LC, JC, SC |
| Peptide, recombinant protein | STA121 dimeric IgA | Materials and methods of this paper | | Protein complex produced from transfected constructs and containing: STA121 HC, STA121 LC, JC |
| Other | CaptureSelect LC-lambda (Mouse) Affinity Matrix | Thermo Fisher RRID:SCR_008452 | Cat#: 194323005 | Affinity matrix for protein purification |
| Other | Superose 6 Increase 10/300 GL | GE Healthcare Life Sciences RRID:SCR_000004 | Cat#: 29091596 | Size exclusion column for protein purification |
| Software, algorithm | Rosetta CryoEM refinement package | *Wang et al., 2016* | RRID:SCR_015701 | |
| Software, algorithm | Phenix | *Afonine et al., 2018a*; *Afonine et al., 2018b* | Phenix RRID:SCR_014224 Phenix.refine RRID:SCR_016736 | |
| Software, algorithm | Pymol Molecular Graphics System | Schrodinger LLC RRID:SCR_014879 | RRID:SCR_000305 | |

*Continued on next page*

*Continued*

| Reagent type (species) or resource | Designation | Source or reference | Identifiers | Additional information |
|---|---|---|---|---|
| Software, algorithm | RELION-3 | *Scheres, 2012*; *Zivanov et al., 2018* | RRID:SCR_016274 | |
| Software, algorithm | UCSF Chimera | *Pettersen et al., 2004*; *Yang et al., 2012* | RRID:SCR_004097 | |
| Software, algorithm | UCSF ChimeraX | *Goddard et al., 2018* | RRID:SCR_015872 | |
| Software, algorithm | cryoSPARC v.2 | *Punjani et al., 2017* | RRID:SCR_016501 | |
| Software, algorithm | VMD | *Humphrey et al., 1996* | RRID:SCR_001820 | |

## Construct design and protein expression

Genes encoding the *Mus musculus* IgA heavy chain constant region (Uniprot P01878) and the lambda light chain constant region (Uniprot A0A0G2JE99) were fused with the STA121 $V_H$ and $V_L$ domain sequences (*Moor et al., 2017*) to create complete heavy chain and light chain sequences. The TPA signal sequence (residues MDAMKRGLCCVLLLCGAVFVSPAGA) was encoded at the start of the heavy chain sequence and the mouse IgKappa signal sequence (residues METD TLLLWVLLLWVPGSTG) was encoded at the start of the light chain sequence. These sequences, along with *Mus musculus* JC (Uniprot P01592; native signal peptide) and *Mus musculus* pIgR ectodomain (SC) residues 1–567 (Uniprot O70570; native signal peptide), were codon optimized, synthesized (Integrated DNA Technologies, Inc) and each cloned into mammalian expression vector pD2610v1 (Atum). Resulting expression constructs were transiently co-transfected into HEK Expi-293-F cells with ExpiFectamine, according to company protocol (Thermo Fisher). Co-transfection to produce SIgA and dIgA utilized equal amounts of each DNA expression construct; SIgA was produced by co-transfecting all four constructs, whereas dIgA was produced by co-transfecting all constructs except for the construct encoding the pIgR ectodomain (SC). Five days following transfection cellular supernatants were harvested and SIgA complexes were purified using CaptureSelect LC-lambda (Mouse) Affinity Matrix (Thermo Fisher) and Superose 6 (GE Healthcare Life Sciences) size exclusion chromotography (SEC). SEC fractions corresponding to the expected size of SIgA (containing two IgA) and dIgA were maintained in buffer containing 20 mM Tris-HCl pH 7.4 and 150 mM NaCl and utilized for cryoEM. Requests for reagents generated in this paper should be directed to the corresponding author, Beth Stadtmueller. Permission to use any reagent containing the STA121 sequence must be obtained from Institute for Research in Biomedicine (Bellinzona, Switzerland).

## CryoEM grid preparation and data collection

Quantifoil R2/2 300 mesh grids were glow discharged using a Pelco easiGlow system for 1 min at 20 mA current. A 3 μL drop of the SIgA or dIgA sample (1.5 mg/ml and 1.0 mg/ml respectively) was applied to the grids and blotted (Whatman #1 paper) for 2 to 8 s using Vitrobot Mark IV (Thermo Fisher) with a blot force of 5, 0 s wait and drain time, at 4°C and 100% RH. Grids were plunged into liquid nitrogen cooled ethane. Movies were collected using SerialEM (*Schorb et al., 2019*) on a Titan Krios (Thermo Fisher) operating at 300 kV, equipped with BioQuantum Energy Filter (20 eV slit width, Gatan) and a K3 direct electron detector (Gatan). SIgA movies were collected at 130,000 magnification in super resolution mode with calibrated pixel size of 0.326 Å/pixel, 40 frames per movie, 0.03 s per frame, and total dose of ~ 60 electrons/$Å^2$. The dIgA movies were collected at 105,000 magnification in super resolution mode with calibrated pixel size of 0.418 Å/pixel, 40 frames per movie, 0.05 s per frame, and total dose of ~ 60 electrons/$Å^2$.

## CryoEM data processing

For the SIgA, 1512 movies were collected. The dataset was processed independently in cryoSparc v. 2.X (*Punjani et al., 2017*) and Relion 3.1-beta. (*Scheres, 2012*; *Zivanov et al., 2018*; *Figure 2—figure supplement 1*, *Figure 2—figure supplement 2*). Maps resulting from the cryoSparc and Relion three processing pipelines were combined (phenix.combine_focused_maps). The resulting

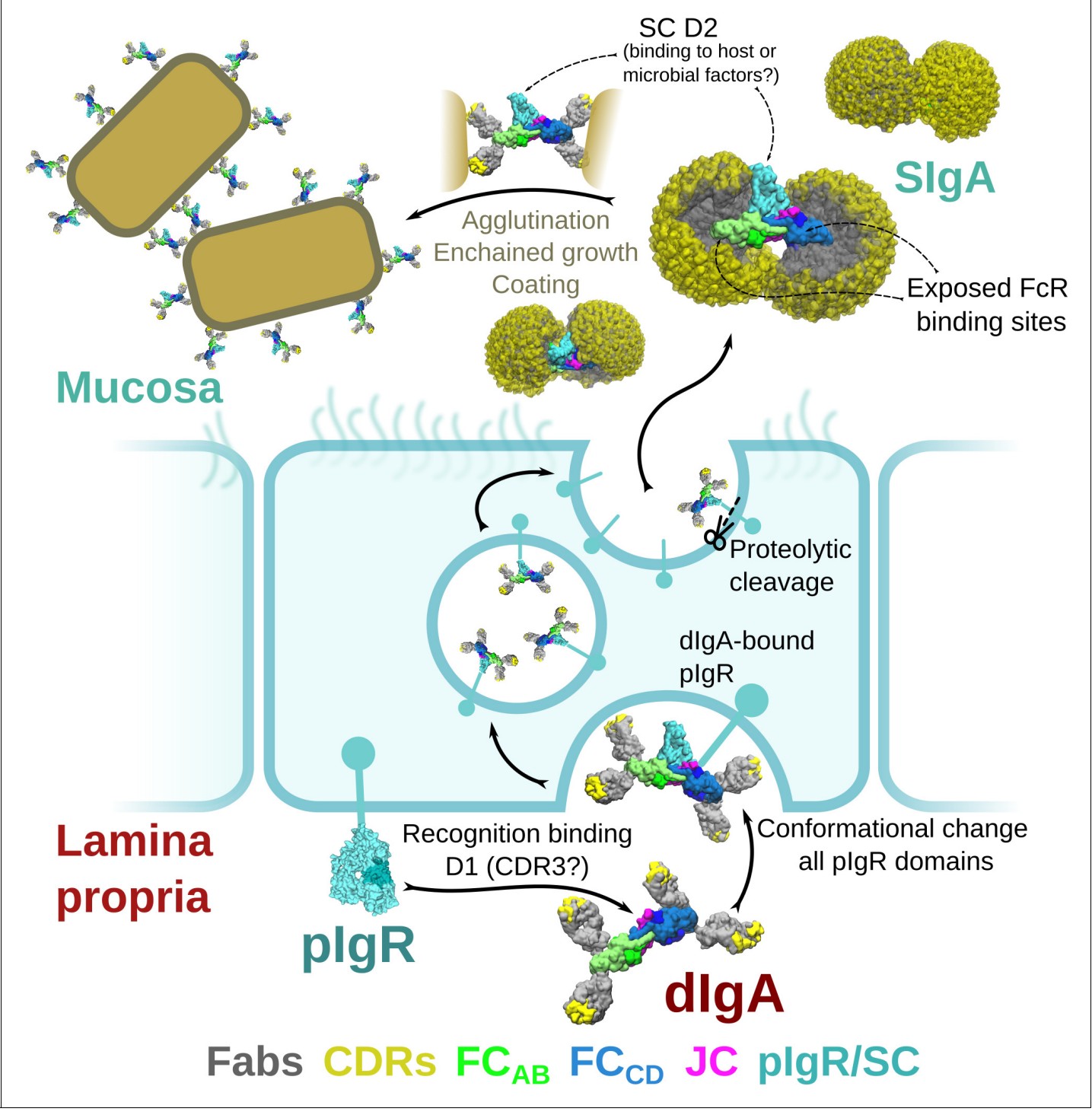

**Figure 7.** Model for the formation, transport, and function of SIgA. Schematic summary depicting the unliganded SC structure (pdb code 5D4K) as pIgR bound to basolateral surface of an epithelial cell in its closed conformation and recognizing bent dIgA from the lamina propria. The pIgR binding to dIgA triggers a conformation change that repositions its domains to facilitate numerous stabilizing contacts with dIgA. The dIgA-pIgR complex transcytoses to the apical membrane where the pIgR is proteolytically cleaved, releasing SIgA into the mucosa. In the mucosa, SIgA Fabs (shown in all possible modeled positions) are directed toward the concave side of the antibody 'looking' for potential antigens while its Fc-receptor-binding regions are exposed on the convex side and accessible to potential host or microbial receptors. SC domains are also partially exposed; the D2 domain is almost completely accessible, protruding out of the SIgA where it may bind host and bacterial factors. Upon encountering antigen, SIgA Fabs bind, promoting antigen coating, agglutination, or enchained growth.

'combined map' was used for initial model building (described below); however, rebuilding and refinement utilized the cryoSparc processing pipeline, which is designated as the primary data in the EMDB.

CryoSparc Live beta was used for the initial processing of the data - movies were motion corrected, binned by 2, and CTF estimation was performed. Particles were initially picked by blob picker with minimum and maximum particle diameter of 100 Å and 120 Å, respectively. Streaming 2D classification was used to generate initial 2D references, which were then used as templates for particle picking. After completion of the collection session, images were screened by limiting total motion to 100 px or less and CTF fit estimation to 10 Å or better, resulting in 1167 selected images or 956,981 picked particles. Particles were exported into cryoSparc for further processing. They were 2D classified into 100 classes and the best classes were selected (623,071 particles). A smaller subset of only large views was also selected (123,533) and four *ab-initio* structures were generated. The best structure (79,904 particles) was refined against the 123,533 particle set to 3.6 Å resolution. The volume was low pass filtered to 60 and both the 3.6 Å map and the filtered volume were used in heterogeneous refinement with the all good particles from 2D classification. The two classes generated a 4.4 Å structure with 368,541 particles and a noise class at 9 Å with 254,530 particles. Good class was refined to 3.4 Å using non-uniform refinement job type. The particles were then used in heterogeneous refinement job type using the output of non-uniform refinement as two input volumes. Once again, one of the classes reached 4.2 Å resolution with 229,252 particles and the other class served as a trap for bad particles (7.22 Å with 139,289 particles). The good particle set was refined using homogeneous refinement job type to 3.5 Å and 3.2 Å with non-uniform refinement.

At the same time, all the movies were motion corrected, binned by 2, and dose-weighted in Relion v.3. CTFs were estimated with CTFFind4 (v. 4.1.13) (*Rohou and Grigorieff, 2015*). Images were screened for ice thickness, contamination, and CTF fits, resulting in 1266 remaining images. Particles were auto-picked with LoG picker in Relion 3 (minimum and maximum diameters of 60 and 180 Å respectively). Approximately 1.4 million particles were extracted, binned four times, and subjected to several rounds of 2D classification. At each step, good classes were selected, and remaining particles were classified again with a smaller mask diameter. Selected particles (310,626) were merged into a single file and re-extracted with 2X binning and further underwent 2D classification. Remaining 184,737 particles were used in a 3D refinement (using an initial model generated in a parallel cryoSparc processing pipeline). After the refinement, particles were re-extracted at full pixel size and refined. After this initial refinement, the particles underwent Bayesian polishing and per-particle-CTF estimation. Particles were once again refined and the map was post-processed and B-factor sharpened, resulting in the final average resolution of 3.7 Å at FSC = 0.143.

For the dIgA, 2951 movies were initially collected using beam-image shift across nine holes in a 3 × 3 pattern. All the processing steps were done in cryoSparc v.2. Movies were imported, motion corrected, and CTFs were estimated. Exposures were then curated to remove bad micrographs, resulting in 2258 remaining images. Particles were picked based on five templates generated from a 4.5 Å structure generated from an earlier, smaller dataset. Particles (1.81 million) were extracted and binned 4X for an initial round of 2D classification. All classes showing secondary structure were selected (688,183). Remaining particles were 2D classified with a smaller circular mask, to try and capture projections along the length of the particle. This round of classification was repeated twice, resulting in additional 168,530 and 59,876 particles. The initial particle subset was used to generate six *ab-initio* structures and then processed with heterogeneous refinement with three classes. The best class was combined with the remaining two subsets and another round of heterogeneous refinement was performed. 462,707 particles belonging to the good class were re-extracted at full pixel size and another round of heterogeneous refinement with three classes was performed. Resulting set of 288,823 particles were refined (Legacy). The particle stack was then split into nine groups based on the pattern of collection. Per-particle CTF refinement, beam tilt, trefoil, and spherical aberration corrections were performed for each group. A final round of homogeneous refinement was performed, resulting in the final overall resolution of 3.3 Å at FSC = 0.143.

## Structure building, refinement, and validation

Starting models for SIgA structure determination were made using SWISS-MODEL (*Waterhouse et al., 2018*). Briefly, homology models of mouse IgA Fc and SC were generated using the amino acid sequences corresponding to individual components, heavy chain and SC, and

reference pdbs files 1OW0 chains A and B (human monomeric IgA Fc) and 5D4K (unliganded human SC), respectively (*Herr et al., 2003*; *Stadtmueller et al., 2016a*). Tps and JC sequences did not exhibit homology to any known structure and were not modeled. Homology models were docked into to real-space electron density using UCSF Chimera (*Pettersen et al., 2004*; *Yang et al., 2012*); each Fcs was docked as a single unit, while SC domains D1, D2, D3, D4, D5 were docked individually. Domain positions were refined as rigid bodies using Phenix (*Afonine et al., 2018b*). Inspection of the map and preliminary SIgA model fit, revealed numerous loops that fit density poorly as well as unaccounted for density at the center of the molecule (Tp and JC). Poorly fitting loops were manually re-built into density using the Coot Molecular Graphics Package (*Emsley and Cowtan, 2004*), along with preliminary placement of a subset of JC and Tp residues. The resulting unrefined SIgA model contained four HC, one SC, and one JC (14 folded domains).

To facilitate building the complete JC and Tp sequences, we implemented a strategy to build and refine the structure using a combination of the Rosetta CryoEM refinement package and Rosetta comparative modeling (RosettaCM) (*Wang et al., 2016*). Briefly, the unrefined starting model and combined map, including data from Relion and Cryosparc processing pipelines, were used to build 2913 Rosetta-modified structures using Rosetta 2019.35.60890, running on a 64 CPU server. All 2913 structures were scored based on geometry (determined by MolProbity score [*Williams et al., 2018*]) and fit, which was approximated with a Rosetta-determined Fourier-shell correlation (FSC) between the structure's calculated map and the experimental density map. We scored each structure by dividing the FSC by the MolProbity score; the four highest-scoring structures were compared to the unrefined SIgA structure and the experimental map. Subsequent iterations of this process were used to rebuild portions of the $C_H2$ and $C_H3$ domains, during which the Phenix phenix.map_-model_cc was used to determine model fit to density. The map surrounding SC D2 was poorly ordered and contained extra density likely representing an ensemble of D2 positions and/or partially ordered carbohydrates attached to the four PNGS on D2. We positioned the domain with the best average fit to the density using distance constraints of the linkers connecting it to D1 and D3. The resulting structure was rebuilt by hand and refined against the map from the cryoSparc processing pipeline using Coot and Phenix (*Emsley and Cowtan, 2004*; *Afonine et al., 2018b*). This process included the addition of carbohydrates at PNGS where well-ordered density for at least one base was visible, as well as refinement of atomic displacement parameters (ADPs).

The dIgA structure was determined using a similar approach as used for the SIgA structure. Briefly, homology models of the mouse IgA Fc were docked into the dIgA map along with the JC from the SIgA structure. Domain positions were refined as rigid bodies using Phenix (*Afonine et al., 2018b*) and poorly fitting regions were re-built by hand. The resulting structure, including carbohydrates, was refined using the Rosetta CryoEM refinement package and Phenix (*Wang et al., 2016*; *Afonine et al., 2018a*).

The final SIgA and dIgA structures and their fit to cryoEM maps were evaluated by hand and validated using Phenix EM Validation, Molprobity, and EMRinger (*Barad et al., 2015*; *Afonine et al., 2018a*; *Williams et al., 2018*); results are summarized in *Supplementary file 1*. Additionally, each final models' fit to unsharpened data was evaluated by calculating map to model FSC curves against all data ($FSC_{map-model}$) and against two half maps, to access overfitting. Briefly, the final refined models were displaced by 0.5 Å after setting the B-factor to 99 using Phenix pdbtools. Each displaced model was refined against half of the data (half map 1) and the resulting refined model and half map 1 were used to calculate a map to model FSC termed '$FSC_{work}$;' the refined model and the half of the data not used in refinement calculation (half map 2) were used to calculate a map to model FSC termed '$FSC_{free}$'.

SIgA and dIgA cryoEM maps and structure coordinate files have been deposited in the EM databank with accession codes EMD-22309 (dIgA) and EMD-22310 (SIgA) and the protein databank with accession codes 7JG1(dIgA) and 7JG2 (SIgA).

## Structure analysis and figures

The sequence alignment between human and mouse counterparts of JC (human: Uniprot P01591, mouse: Uniprot P01592), pIgR ectomain (human: Uniprot P01833, mouse: Uniprot O70570) and heavy chain- $C_H2$-$C_H3$ regions (human IgA1: Uniprot P01876, human IgA2: Uniprot P01877, human IgA two allotypes [*Lombana et al., 2019*], mouse IgA: Uniprot Q99LA6) were carried out using

ClustalOmega (*Sievers et al., 2011*) and figures were made with EsPript 3 (*Gouet et al., 1999*). Percent Identity and similarity values were determined using NCBI-BLAST (*Altschul et al., 1990*).

Contacts between individual Tps and the JC and all other SIgA components were evaluated by inspecting all interfacing residues within approximately 7 Å. The probability of conserved contacts was evaluated by hand inspection of the cryoEM map. A list of all possible interactions between SC and the components of dIgA were made by using the Protein Interaction Calculator (PIC) webserver (*Tina et al., 2007*). The list of interactions were visualized in Pymol Molecular Graphics System (Schrodinger LLC) to validate the distances and additional unspecified interactions within 4A. The PDBePISA server (https://www.ebi.ac.uk/pdbe/pisa/) (*Krissinel and Henrick, 2007*) was used for the calculation of interface surface area, between various elements of the structure. Pymol Molecular Graphics System (Schrodinger LLC) was used to calculate the percentage of accessible surface area of in SIgA and dIgA. The potential N-linked glycosylation sites were determined using the NetNGlyc 1.0 Server (*Blom et al., 2004*). The centroid axes and centroid planes for $Fc_{AB}$ and $Fc_{CD}$ (residues 237–445) were individually determined using UCSF Chimera. The angles between the axes and planes were then measured relative to one another from the Tools > analysis > centroid/axes/planes feature (*Pettersen et al., 2004*; *Yang et al., 2012*). The RMSD differences between SIgA and dIgA structures were calculated using PyMol. To visualize the difference between SIgA and dIgA, structures were aligned on JC Cα atoms, which have a RMSD of 0.891. Figures were made using the Pymol Molecular Graphics System (Schrodinger LLC), UCSF Chimera (*Pettersen et al., 2004*; *Yang et al., 2012*), UCSF ChimeraX (*Goddard et al., 2018*) and Inkscape (https://inkscape.org/).

## Computational search for potential Fab positions

The pdbs files used in Fab modeling were generated using the template mode of SWISS-MODEL (*Waterhouse et al., 2018*) with reference PDB file 4EOW chain A and chain B and the STA121 $C_H1$-$V_H1$ and $C_L1$-$V_L1$ sequences (*Moor et al., 2017*), respectively. In order to evenly sample all the potential positions, each Fab was rotated such that the center of mass (C.O.M) of the CDR was arranged on a Fibonacci spherical lattice (FSL) of one thousand points (*Marques et al., 2013*). During the rotation, the N-terminal residue in the $C_H2$ domain (pivot) was set to be in the center of the FSL. Each of 1000 generated orientations was further rotated along the axis spanning from the pivot to the CDR, with an interval of γ = 45°. In total, 8000 structures were generated for each Fab. Any position in which two or more Fab amino acids had a clash was removed from the data set. Clashes were defined if more than 8 atoms of Fab (except the linker) were within 1 Å of any atom of the Fc, JC and SC domains of SIgA. A minimum of 8 atoms was chosen to define a clash because one amino acid has on average eight heavy atoms. The linker region of the Fab was not considered for the clash calculation because we assumed that the linker is flexible. The number of positions sampled by Fab CDRs were quantified by measuring φ, defined to be the angle between the Fab vector and the y-axis, a vector parallel to the $Fc_{AB}$ plane and passing through the center of mass of Fc and JC. Figure were made using VMD (*Humphrey et al., 1996*).

## Acknowledgements

We thank Pamela Bjorkman and members of the Bjorkman lab at Caltech for supporting preliminary work related to this study as well as Jost Vielmetter and the Caltech Protein Expression Center (housed and funded in part by the Caltech Beckman Institute) for guidance on IgA protein expression strategies. We also thank members of the Stadtmueller Laboratory, as well as Emma Slack (ETH Zurich), for insightful conversations regarding the SIgA and dIgA structures. Cryo Electron microscopy was done in the Beckman Institute cryo-EM resource center at Caltech. The STA121 antibody sequence was provided by Luca Piccoli and the Institute for Research in Biomedicine (Bellinzona, Switzerland). This study was funded by University of Illinois Urbana-Champaign start-up funding, National Institutes Health (United States) grant P41-GM10460, and National Institutes Health (United States) grant R01 AI041239. Molecular graphics and analyses performed with UCSF Chimera, developed by the Resource for Biocomputing, Visualization, and Informatics at the University of California, San Francisco, with support from NIH P41-GM103311. Molecular graphics and analyses performed with UCSF ChimeraX, developed by the Resource for Biocomputing, Visualization, and Informatics at the University of California, San Francisco, with support from National Institutes of Health R01-

GM129325 and the Office of Cyber Infrastructure and Computational Biology, National Institute of Allergy and Infectious Diseases.

## Additional information

### Funding

| Funder | Grant reference number | Author |
|---|---|---|
| University of Illinois at Urbana-Champaign | Start-up funding | Sonya Kumar Bharathkar Benjamin W Parker Beth Stadtmueller |
| National Institute of General Medical Sciences | P41-GM10460 | Nandan Haloi Emad Tajkhorshid |
| National Institutes of Allergy and Infectious Diseases | R01 AI041239 | Kathryn E Huey-Tubman Beth Stadtmueller |

The funders had no role in study design, data collection and interpretation, or the decision to submit the work for publication.

### Author contributions

Sonya Kumar Bharathkar, Benjamin W Parker, Conceptualization, Resources, Formal analysis, Validation, Investigation, Visualization, Methodology, Writing - original draft, Writing - review and editing; Andrey G Malyutin, Validation, Investigation, Visualization, Writing - original draft; Nandan Haloi, Formal analysis, Visualization, Methodology, Writing - original draft; Kathryn E Huey-Tubman, Resources, Methodology, Writing - review and editing; Emad Tajkhorshid, Supervision, Funding acquisition, Methodology; Beth M Stadtmueller, Conceptualization, Resources, Formal analysis, Supervision, Funding acquisition, Validation, Investigation, Visualization, Writing - original draft, Project administration, Writing - review and editing

### Author ORCIDs

Sonya Kumar Bharathkar https://orcid.org/0000-0002-3637-4990
Benjamin W Parker https://orcid.org/0000-0002-1464-1189
Emad Tajkhorshid http://orcid.org/0000-0001-8434-1010
Beth M Stadtmueller https://orcid.org/0000-0003-0637-3206

### Decision letter and Author response

Decision letter https://doi.org/10.7554/eLife.56098.sa1
Author response https://doi.org/10.7554/eLife.56098.sa2

## Additional files

### Supplementary files

• Supplementary file 1. CryoEM data collection and refinement statistics associated with SIgA and dIgA structures.

• Transparent reporting form

### Data availability

SIgA and dIgA cryoEM maps and structure coordinate files have been deposited in the EM databank with accession codes EMD-22309 (dIgA) and EMD-22310 (SIgA) and the protein databank with accession codes 7JG1(dIgA) and 7JG2 (SIgA).

The following datasets were generated:

| Author(s) | Year | Dataset title | Dataset URL | Database and Identifier |
|---|---|---|---|---|
| Bharathkar SK, Par- | 2020 | Secretory Immunoglobin A (SIgA) | https://www.rcsb.org/ | RCSB Protein Data |

| | | | | | |
|---|---|---|---|---|---|
| ker BW, Malyutin AG, Stadtmueller BM | | | | structure/7JG2 | Bank, 7JG2 |
| Bharathkar SK, Parker BW, Malyutin AG, Stadtmueller BM | 2020 | Dimeric Immunoglobin A (dIgA) | | https://www.rcsb.org/structure/7JG1 | RCSB Protein Data Bank, 7JG1 |
| Bharathkar SK, Parker BW, Malyutin AG, Stadtmueller B | 2020 | Dimeric Immunoglobin A (dIgA) | | https://www.emdataresource.org/EMD-22309 | EMDataResource, EMD-22309 |
| Bharathkar SK, Parker BW, Malyutin AG, Stadtmueller B | 2020 | Secretory Immunoglobin A (SIgA) | | https://www.emdataresource.org/EMD-22310 | EMDataResource, EMD-22310 |

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
