## [Decision Letter]

Thank you for submitting your article "The Structures of Secretory and Dimeric Immunoglobulin A" for consideration by *eLife*. Your article has been reviewed favorably by two peer reviewers, including Sjors HW Scheres as the Reviewing Editor and Reviewer #1, and the evaluation has been overseen by John Kuriyan as the Senior Editor. The following individual involved in review of your submission has agreed to reveal their identity: Martin F Flajnik (Reviewer #2).

The reviewers have discussed the reviews with one another and the Reviewing Editor has drafted this decision to help you prepare a revised submission.

We would like to draw your attention to changes in our revision policy that we have made in response to COVID-19 (https://elifesciences.org/articles/57162). Specifically, we are asking editors to accept without delay manuscripts, like yours, that they judge can stand as *eLife* papers without additional data, even if they feel that they would make the manuscript stronger. Thus, most of the revisions requested below only address clarity and presentation. Note, however, that one of the points made below (7) concerns a suggestion that additional refinement be done on the structures. Hopefully this can be done quickly under the present circumstances. If not, please explain why when submitting the revised manuscript.

Summary:

This paper describes structures of secretory immunoglobulin A (IgA) and its dimeric form. The field has waited a long time for the structure of these complexes. The senior author of this paper has examined interaction of pIgR with Igs previously, and now has the hard data to probe existing models. The paper is straightforward, providing important data on how the secretory component and the J chain modify the IgA Fc structure, which in turn answers most of the questions that have dogged the field for many years. No new experiments are suggested, but some features should be discussed better, as outlined below.

Essential revisions:

1) It has always been mysterious 'where the J chain came from.' Are there indications from the structure as to homologies that might have been missed because of low sequence similarity to J chain relatives?

2) The J chain is very acidic; specialized types of gels must be run so that the J chain is separated from Ig light chains (13 kd vs. 25 kd, respectively), as the high acidic nature of the J chain is thought to repel SDS on conventional gels. Why is it so acidic-does it make many ionic bonds?

3) Bony fish have lost the J chain in evolution, but still have a pIgR transporter with D1/D5. Any ideas of how teleost IgM can interact with teleost pIgR for transport without J chain? The senior author has commented on this in one of her previous papers, but now may have better ideas.

4) Any idea how monomeric IgA could "work" now that we have the complete dimeric structure? Mice only have one IgA gene, so how would (could) it function without the J chain. Also in humans, one of the two IgA genes seems to preferentially associate with the J chain. Any ideas on this topic now that we know the structure of the mouse dimeric IgA?

5) Perhaps the authors could briefly comment on how this work confirms or is different from the two recent reports in Science on the same/similar topic.

6) Much of the text, but especially the Results section, is written in a very dry and descriptive manner, which makes it hard to follow. Perhaps the authors could remove some of the detailed mentions of structural features and to describe their results in a more accessible manner. For example, instead of mentioning lots of individual contacts with their detailed residue numbers, the authors could try to convey to the general reader of *eLife* what these interactions mean.

7) Atomic modelling has been done in a non-conventional manner. The authors claim this was done to "counter challenges of hand building five chains in a moderate resolution map". However, at 3.3A and 3.7A that should have been not much of a problem. Anyway, it doesn't really matter how the initial model was generated (the authors used Rosetta), but at these resolutions the final models should be properly refined into the cryo-EM density map. The authors could choose refmac or phenix (without Ramachandran restraints if the authors wish to use these for validation) to easily do so. They should then also present FSC curves between the final model and the cryo-EM map, together with FSC_free_ and FSC_work_ curves, i.e. FSC model-vs.-map from a model refinement in one of the half-maps against that same half-map and of that same model against the other half-map, to assess overfitting. Also, they should make detailed zoomed-in supplementary figures that clearly illustrate the quality of the atomic model fit and of the cryo-EM fit in different regions of the complex.

8) The cryo-EM map calculation seems to have been done to an acceptable standard, but some details are missing from the manuscript:

– Please include pictures of one original micrograph for both structures.

– How were local resolution estimates in Figures 2—figure supplement 1 and Figure 5—figure supplement 1 obtained?

---

## [Author Response]

Essential revisions:1) It has always been mysterious 'where the J chain came from.' Are there indications from the structure as to homologies that might have been missed because of low sequence similarity to J chain relatives?

Indeed, the question of where the J chain (JC) came from is an important, yet outstanding, question. To address this question, we have searched the PDB for proteins and complexes that are structurally similar to the J-chain and/or the J-chain/tailpiece assembly; however, we have not identified strong candidates. We have noted the results of this analysis in the revised Discussion section, in which the second paragraph is now devoted to the JC. We anticipate that future efforts to determine the structure of the unliganded J-chain may reveal an alternative JC conformation (not stabilized by interactions with the IgA heavy chains) and could provide additional insight on the evolution of JC and its relationship to other proteins.

2) The J chain is very acidic; specialized types of gels must be run so that the J chain is separated from Ig light chains (13 kd vs. 25 kd, respectively), as the high acidic nature of the J chain is thought to repel SDS on conventional gels. Why is it so acidic-does it make many ionic bonds?

The mouse J chain (JC) sequence has many acidic residues; however, our analysis does not point to a single, defined structural role for the majority of JC acidic residues. We do find that the JC uses acidic residues to bind secretory component (SC) and have noted this more clearly in the revised Results section. Yet, the JC-SC interface is minimal compared to interfaces between JC and IgA heavy chains and JC acidic residues do not appear to dominate these interfaces. JC acidic residues otherwise appear to be broadly distributed on exposed surfaces. The structure of unliganded JC remains unknown and thus, it is possible that acidic residues play some role in stabilizing the unliganded structure and/or contribute to the JC-IgA binding mechanism; yet without additional data, we can only speculate on this topic.

3) Bony fish have lost the J chain in evolution, but still have a pIgR transporter with D1/D5. Any ideas of how teleost IgM can interact with teleost pIgR for transport without J chain? The senior author has commented on this in one of her previous papers, but now may have better ideas.

Our SIgA structure reveals numerous contacts between the IgA heavy chains and SC D1 and D5, raising the question of whether or not analogous interactions between teleost pIgR (SC) domains and teleost IgM or IgT (a teleost mucosal antibody), might be sufficient for binding in the absence of the JC. This is a fascinating question; however, our structural analysis has not provided a meaningful answer. Teleost pIgR has just two domains with reported homology to mammalian SC D1 and D5; however, D1 and D5 residues involved in contacts with IgA are not well conserved in teleost SC domains (Stadtmueller et al., 2016). Furthermore, in the absence of teleost IgM and/or IgT structures, our ability to predict how teleost D1/D5 might engage its antibody ligands remains limited and beyond what we can reliably discuss in this paper.

4) Any idea how monomeric IgA could "work" now that we have the complete dimeric structure? Mice only have one IgA gene, so how would (could) it function without the J chain. Also in humans, one of the two IgA genes seems to preferentially associate with the J chain. Any ideas on this topic now that we know the structure of the mouse dimeric IgA?

IgA can also function as a monomer, the form typically found in serum (Woof and Russell, 2011). Our mouse SIgA and dIgA structures reveal beta-sheet interactions between the tailpieces (Tps) from two IgA monomers, interactions that might be able to link two antibodies in the absence of J chain (JC). However, the existence of monomeric IgA in vivo, and our ability to express monomeric mouse IgA in transiently transfected HEK cells in the absence of the JC (data not shown), implies that interactions between Tps from two or more IgA monomers do not occur, or are not stable, in the absence of the JC. One plausible mechanism that might account for this is that the Tps adopt alternative conformations in the absence of JC and/or that JC interactions with the IgA heavy chain induce conformational change(s) that promote beta-sheet like interactions between Tps. Although additional experiments will be needed to test these hypotheses, we have revised the manuscript to include discussion on potential mechanisms of polymeric antibody assembly and how the JC and the IgA Tps may contribute to that process.

The human genome encodes two IgA isotypes, IgA1 and IgA2 (and two IgA2 allotypes IgA2m1 and IgA2m2), whereas the mouse genome encodes a single IgA. Although the majority monomeric serum IgA is IgA1, all of the human isotypes and allotypes can form polymeric antibodies with JC and the relative abundance of each type differs in each mucosal environment (e.g. nasopharynx, gut, urogenital track, etc.) (Woof and Russell, 2011). As noted in our revised Discussion, sequence conservation and structural similarities among mouse dIgA, SIgA and concurrently reported structures of the Fc region of human SIgA1 (dimeric form) suggest a conserved SIgA core structure in dimeric forms of IgA among mammals. Our analysis reveals a handful of sequence differences in residues likely to be bridging interactions between JC and the IgA heavy chains in mouse and human SIgAs; yet, it is not immediately clear from structural data alone if and how these differences might uniquely affect JC association with specific IgA heavy chains. It is possible that the limited sequence variability found among IgA heavy chains alters the dimeric IgA assembly process in some way; however other factors are likely to influence this process. For example, recent reports indicate that in transiently transfected mammalian cells, the ratio of transfected JC DNA to IgA heavy chain DNA influences the polymeric state of resulting IgAs (Lombana et al., 2019); molecular chaperones have also been recently implicated in IgA assembly (Suzuki, Vogelzang and Fagarasan, 2019, Xiong et al., 2019). Further studies will be needed to define the molecular mechanism of polymerization and to hammer out the roles that the JC and each unique IgA heavy chain sequence play in this process.

5) Perhaps the authors could briefly comment on how this work confirms or is different from the two recent reports in Science on the same/similar topic.

The preprint of this manuscript (https://www.biorxiv.org/content/10.1101/2020.02.16.951780v1) was posted February 17, 2020, several days after the structures of the Fc regions of human dimeric SIgA1, tetrameric and pentameric SIgA2 were reported as an advanced online publication in Science (and subsequently appeared in the February 28^th^ 2020 issue of Science). Given this, we have focused our manuscript on the original work we performed rather than a detailed comparison between our structures and human IgA structures, which we believe is better suited for a review. As noted below, we have included some simple comparisons between the structures for the purpose of discussing conservation of SIgA structures.

Our work differs from structures reported in Science in several ways:

1) We report mouse, rather than human SIgA structures. As described in our revised manuscript, for the purpose of discussing conservation of SIgA structure among mammals we have completed a limited comparison of the overall conformation of the mouse SIgA and human dimeric SIgA1 Fc, which reveals largely superimposable structures. We do note differences in some interface residues; however, without additional experiments it is difficult to evaluate the significance of these findings. Together, our observations suggest a conserved conformation at the core of SIgAs released into the mucosa and suggest that with respect to SIgA structure, mouse model systems are relevant to understanding human SIgA structure and functions.

2) In addition to SIgA, we report the structure of dIgA (lacking secretory component). The dIgA structure reveals that dimeric forms of IgA adopt a conformation characterized by the same angle of bend as observed for SIgA. This points toward dIgA having limited conformational flexibility and to a dominant role for the JC and tailpieces in maintaining the bent conformation of the two IgA Fcs. Furthermore, differences between tilt in dIgA and SIgA and signify that SC can influence the conformation (tilt) of SIgA.

3) We report modeling which reveals how the conformation of SIgA (bend and tilt) could influence the positions of Fabs and thus, interactions with antigen. These results are an important step toward understanding the mechanisms by which SIgAs can neutralize pathogens and interact with commensal microbes.

4) Our methods for sample preparation are different than those reported in Science. For example, in contrast to the human SIgA core structures which lacked fabs and reportedly utilized crosslinking reagents to stabilize protein on cryoEM grids, the complexes we used for studies were intact SIgAs (and dIgAs) with Fabs. These complexes were also prepared for cryoEM in the absence of crosslinking reagents. Taken together, it is clear that multiple methods can be successful for structural studies of IgA.

6) Much of the text, but especially the Results section, is written in a very dry and descriptive manner, which makes it hard to follow. Perhaps the authors could remove some of the detailed mentions of structural features and to describe their results in a more accessible manner. For example, instead of mentioning lots of individual contacts with their detailed residue numbers, the authors could try to convey to the general reader of eLife what these interactions mean.

We thank the reviewers for this suggestion. We have re-written the Introduction and Results sections in an effort to make the information more accessible to a broad audience. These modifications include minimal changes to main text Figures 3 and 4. We have also modified Figure 1—figure supplement 1 to show a more detailed sequence alignment and report similarity and identity between human and mouse SIgA component sequences. We have also added Figure 2—figure supplement 4, showing the locations of modeled carbohydrates on the SIgA structures. As noted below, we have also updated supplementary figures to detail more of the cryoEM pipeline.

7) Atomic modelling has been done in a non-conventional manner. The authors claim this was done to "counter challenges of hand building five chains in a moderate resolution map". However, at 3.3A and 3.7A that should have been not much of a problem. Anyway, it doesn't really matter how the initial model was generated (the authors used Rosetta), but at these resolutions the final models should be properly refined into the cryo-EM density map. The authors could choose refmac or phenix (without Ramachandran restraints if the authors wish to use these for validation) to easily do so. They should then also present FSC curves between the final model and the cryo-EM map, together with FSC_free_ and FSC_work_ curves, i.e. FSC model-vs.-map from a model refinement in one of the half-maps against that same half-map and of that same model against the other half-map, to assess overfitting. Also, they should make detailed zoomed-in supplementary figures that clearly illustrate the quality of the atomic model fit and of the cryo-EM fit in different regions of the complex.

The SIgA map was not well resolved in loop regions connecting secondary structure elements within the Tps and JC, making it difficult to assign sequence for five chains in the absence of other structural data (there were not previously determined structures or predictions of the fold for JC or the IgA Tp) and thus we found a combination of hand building and Rosetta modeling that utilized cryoEM data, to be an efficient strategy which may be useful for determining other challenging structures at moderate resolutions.

As noted in our Materials and methods, we utilized a published Rosetta cryoEM pipeline which included refinement against cryoEM data and subsequently refined the structures using Phenix refine. We have modified the wording in the Materials and methods sections to make this clear.

We thank the reviewer for suggesting additional figures to assess the model to map fit. Our revised manuscript includes FSC curves between the final model and the cryo-EM map, FSC_free_ and FSC_work_ curves (Figure 2—figure supplement 3, Figure 5—figure supplement 2) and associated methods used to calculate these curves.

8) The cryo-EM map calculation seems to have been done to an acceptable standard, but some details are missing from the manuscript:- Please include pictures of one original micrograph for both structures.- How were local resolution estimates in Figures —figure supplement 1 and Figure 5—figure supplement 1 obtained?

Figure 2—figure supplement 1 and Figure 5—figure supplement 1 have been updated.

In our revised manuscript, we have clarified the methods related to cryoEM map calculations, included figures of micrographs, and described how local resolution maps were calculated in the associated figure legends. Briefly, local resolution maps were calculated using cryoSparc and were prepared in ChimeraX; the scale bar was generated in Chimera.

Modifications to our cryoEM methods also include clear description of two independent SIgA data processing pipelines that utilized cryoSparc (Figure 2—figure supplement 1) and Relion 3 (Figure 2—figure supplement 2). These two processing pipelines resulted in cryoEM maps with average resolutions of 3.7Å and 3.3Å at FSC=0.143, respectively. Initially, we merged Relion 3 and cryoSparc maps using the Phenix combine maps utility (see revised Materials and methods) resulting in a “combined map” that was used for initial Rosetta-assisted building and refinement. After initial model building and careful inspection of the combined, Relion and CryoSparc maps, we utilized the 3.3Å cryoSparc map for the final stages of model building and refinement. It should be noted that refinement of the final structure against the combined map and the cryoSparc map produced the nearly identical statistics; however, the EMDB requires a single map to be designated as primary (cryoSparc map). In sum, we felt that to maintain transparency of approach we should report both processing pipelines that contributed to structure determination.